# Vitamin C Suppresses Pancreatic Carcinogenesis through the Inhibition of Both Glucose Metabolism and Wnt Signaling

**DOI:** 10.3390/ijms232012249

**Published:** 2022-10-14

**Authors:** Ji Hye Kim, Sein Hwang, Ji-Hye Lee, Se Seul Im, Jaekyoung Son

**Affiliations:** Department of Biomedical Sciences, Asan Medical Center, AMIST, University of Ulsan College of Medicine, Seoul 05505, Korea

**Keywords:** Vitamin C, PDAC, glucose metabolism, Wnt signaling, metastasis

## Abstract

Cumulative studies have indicated that high-dose vitamin C has antitumor effects against a variety of cancers. However, the molecular mechanisms underlying these inhibitory effects against tumorigenesis and metastasis, particularly in relation to pancreatic cancer, are unclear. Here, we report that vitamin C at high concentrations impairs the growth and survival of pancreatic ductal adenocarcinoma (PDAC) cells by inhibiting glucose metabolism. Vitamin C was also found to trigger apoptosis in a caspase-independent manner. We further demonstrate that it suppresses the invasion and metastasis of PDAC cells by inhibiting the Wnt/β-catenin-mediated epithelial-mesenchymal transition (EMT). Taken together, our results suggest that vitamin C has therapeutic effects against pancreatic cancer.

## 1. Introduction

Pancreatic ductal adenocarcinoma cells (PDACs) are one of the most aggressive and lethal known tumors and account for more than 90% of all pancreatic malignancies. PDACs show significant resistance to most conventional treatments, including chemotherapy and radiotherapy [1,2]. As a result, the 5-year survival rate of affected patients is less than 8%. Accumulating evidence has now demonstrated that the high mortality associated with PDAC is predominantly caused by the invasiveness of these lesions, resulting in metastases to distant organs such as the liver, lung, and lymph nodes [3,4,5].

Reprogrammed metabolism has been recognized as a major driver of PDAC tumorigenesis. Previous studies have reported that this process is regulated by various oncogenic signals. The KRAS mutation can promote the glycolytic pathway by upregulating the expression of glucose transporters and glycolytic enzymes [6]. Glycolysis is the major carbon metabolic pathway that provides adenosine triphosphate (ATP) production, fuel cell growth and biomass. Furthermore, the enhanced glycolysis associated with PDAC supports the requirements for distant metastasis to cells, the division, invasion, and migration of tumor cells [7,8].

The epithelial-mesenchymal transition (EMT) is a key initiating step in PDAC progression. EMT is regulated by transcription factor expression and leads to a conversion of epithelial cells into more aggressive mesenchymal-like cells that have increased motility [9]. The EMT processes are controlled by the regulation of EMT transcriptional factors such as Snail, Twist, and Zeb. Snail is the primary control factor for the EMT process. It binds to an E-box site in the promoter of the E-cadherin gene that inhibits its transcription and thereby triggers the EMT pathway [10]. Glycogen synthase kinase-3 (GSK-3) is also one of the regulatory factors in the EMT process and acts as a tumor suppressor via the phosphorylation of β-catenin. GSK-3 induces the ubiquitination and degradation of β-catenin and Snail, leading to the inhibition of the Wnt/β-catenin pathway [11]. In addition, cumulative evidence has indicated that high-dose vitamin C inhibits the EMT pathway in various cancer cell types. It has been reported, for example, that high-dose vitamin C inhibits TGF-β1-induced invasion and migration by inhibiting EMT in breast cancer cells [12]. Vitamin C also activates epigenetic modifications through 5-aza treatment in HCC cell lines. The combination of epigenetic regulation and vitamin C downregulates Snail expression, leading to EMT inhibition [13]. Given that Wnt/β-catenin signaling plays an important role in Snail expression, we hypothesized that vitamin C might regulate the expression of Snail-mediated EMT markers to inhibit PDAC cell metastasis by regulating Wnt signaling.

Vitamin C is a natural compound that plays an important role in numerous physiological processes, such as antiviral responses, collagen synthesis and antioxidant pathways [14,15]. Notably, also, previous reports have further indicated that a high dose of vitamin C can also promote the apoptotic death of tumor cells [16]. The pharmacological concentration of vitamin C (0.05–3.53 mg/mL) has been found to induce cell death in various cancer cell lines and not in normal cells. Moreover, because a high dose of vitamin C acts as a pro-oxidant, it causes the accumulation of hydrogen peroxide in plasma, leading to the production of reactive oxygen species (ROS) [17]. Additionally, high-dose vitamin C has been found to improve the sensitivity of current cancer treatments by regulating metabolic aberrance, including a reduction in ATP and glucose transporter 1 (GLUT1) levels [18].

In our present study, we demonstrate that vitamin C has inhibitory effects against PDAC cell growth and metastasis. We found from our experiments that vitamin C exposure suppresses growth and induces caspase-independent apoptotic cell death in PDAC cells by inhibiting glucose metabolism. Interestingly, we also observed that vitamin C induces a significant reduction in the Snail-mediated mesenchymal-marker level, leading to an inhibition of metastasis by PDAC cells. Hence, our current findings suggest that vitamin C administration is a possible new therapeutic approach to PDAC treatment.

## 2. Results

### 2.1. Vitamin C Impairs Pancreatic Cancer Growth

Previous reports have shown that vitamin C has anticancer effects against various tumor types [19,20]. We first explored its effects on PDAC growth. To address this, we treated PDAC cells with vitamin C in PDAC cells and assayed them for cell growth. As shown in Figure 1A and Appendix A, vitamin C treatment significantly inhibited PDAC cell growth. To confirm this effect, we also assayed for cell viability and found consistently that the proliferation and colony formation of PDAC cells was also markedly reduced in the presence of vitamin C (Figure 1B,C). Therefore, these results suggested that vitamin C has an anticancer effect against pancreatic cancer. 

### 2.2. Vitamin C Induces Apoptotic Cell Death in a Caspase-Independent Manner

To further explore the inhibitory effects of vitamin C on PDAC cell survival, we performed an annexin V/propidium iodide (PI) assay in order to examine the effect of vitamin C on cell death in PDAC cells. As shown in Figure 2A and Appendix A, vitamin C treatment (4 or 5 mM) of these cultures triggered apoptotic cell death in a dose-dependent manner. We would like to use the concentration of vitamin C that affects both growth and cell death. Thus, we used 4 or 5 mM of vitamin C in order to explore the antitumor effects in PDAC cells. In general, apoptotic cell death is mediated via either caspase-dependent or -independent pathways. We thus tested the protein level of the effectors caspase-3 and poly-(ADP-ribose) polymerase (PARP) to investigate whether the apoptotic cell death caused by vitamin C is caspase-dependent. Vitamin C did not induce the cleavage of caspase-3 or PARP (Figure 2B). Furthermore, the apoptotic cell death induced by vitamin C was not inhibited by benzyloxycarbonyl-Val-Ala-Asp-(OMe) fluoromethyl ketone (zVAD-fmk), a known pan-caspase inhibitor (Figure 2C). Taken together, these results demonstrated that vitamin C has antitumor effects on PDAC cell survival.

### 2.3. Vitamin C Exerts Antitumor Effects on PDAC Cells by Inhibiting Glucose Metabolism

One of the distinctive features of proliferating cancer cells is enhanced glucose metabolism, reflecting the fact that they have a higher metabolic demand for supply cell components and an enhanced glycolytic flux that produces ATP [8]. A previous report confirmed that overexpression of GLUT1 by a KRAS/BRAF mutation elevates vitamin C uptake. Increased vitamin C uptake induces oxidative stress, leading to the inhibition of glycolysis through GAPDH inactivation, resulting in ATP depletion and cell death [21]. We thus investigated the functional roles of vitamin C in PDAC metabolism. As shown in Figure 3A and Appendix A, the ATP levels were significantly reduced in the presence of vitamin C. To further assess the functional role of vitamin C in glucose metabolism, the extracellular acidification rate (ECAR) was measured and found to be decreased by this treatment (Figure 3B and Appendix A).

We further investigated the effects of vitamin C on glucose metabolism using targeted liquid chromatography-tandem mass spectrometry (LC-MS/MS). The levels of glycolysis intermediates were clearly reduced by vitamin C treatment (Figure 3C and Appendix A), further confirming that vitamin C inhibits glucose metabolism. Previous studies have reported that cancer cells undergo metabolic reprogramming to replenish their survival needs when glycolysis is suppressed, including toward mitochondrial OXPHOS [22,23]. Based on these studies, we attempted to rescue glucose metabolism inhibition-induced cell death by supplementing the cells with pyruvate, which provides substrates for the TCA cycle. As shown in Figure 3D and Appendix A, we tested and found that the suppression of PDAC cell growth by vitamin C treatment was significantly recovered by pyruvate supplementation. Consistently, the inhibition of colony formation by vitamin C was also rescued by the addition of pyruvate to the cultures (Figure 3E and Appendix A), as was apoptotic cell death (Figure 3F and Appendix A). Our analyses thus confirmed that vitamin C induces cell growth inhibition and cell death through the suppression of glucose metabolism. 

### 2.4. Vitamin C Suppresses the Migration Ability of PDAC Cells

Metastases are a hallmark of severe and aggressive cancers and are responsible for about 90% of cancer-related deaths. Many studies have been conducted to date, therefore, to identify treatments that can target metastasis [24]. The EMT plays an important role in cancer metastasis and is thus one of the more attractive targets for developing new cancer treatments [25,26,27]. A previous report has indicated that high-dose vitamin C exposure suppresses the invasion and migration of breast cancer cells through the regulation of EMT marker expression [12]. Hence, to investigate the impact of vitamin C in PDAC EMT in our present study, we explored the expression levels of EMT markers in PDAC cells. We found that vitamin C exposure reduced N-cadherin and Vimentin but increased Zo-1 and E-cadherin expression in PDAC cells in a dose-dependent manner (Figure 4A). We next tested the invasion and migration ability of the PDAC cells in the presence of vitamin C. A Transwell chamber assay revealed that vitamin C treatment significantly inhibited the invasiveness and migratory ability of these cells (Figure 4B,C).

The functional loss of E-cadherin is one of the hallmarks of EMT, and this factor has a critical role as a suppressor of metastasis. E-cadherin was found in prior studies to be repressed by several transcription factors, such as the Snail/Slug family, Twist, ZEB1 and SIP1. Snail is the most important of the transcriptional repressors of E-cadherin [28,29]. We speculated, therefore, that vitamin C may control Snail expression. As shown in Figure 4D, vitamin C treatment of PDAC cells suppressed their Snail levels. To further confirm that Snail plays an important role in vitamin C-mediated EMT, we tested and then observed that we could rescue the effects of a vitamin C-mediated EMT reduction by overexpressing Snail (Figure 4E). Taken together, these results suggest that vitamin C affects the expression of Snail-mediated EMT markers, leading to the inhibition of PDAC cell mobility. 

### 2.5. Vitamin C Inhibits the Migration Ability of PDAC Cells by Inhibiting Wnt/β-Catenin Signaling

Our present results showed that vitamin C regulates Snail-mediated EMT marker expression. The Wnt/β-catenin signaling pathway is known as the canonical Wnt signaling mechanism responsible for the abnormal regulation of the β-catenin. During EMT, β-catenin is localized and accumulated in the nucleus, where it interacts with T cell-specific factor (TCF)/Lymphoid enhancer-binding factor (LEF) to mediate the transactivation of target genes, including EMT markers [30,31]. We thus speculated that vitamin C might regulate Wnt/β-catenin signaling and tested this using a TOP/FOP assay. As shown in Figure 5A, vitamin C treatment indeed significantly inhibited Wnt/β-catenin activity. In addition, the non-phospho-β-catenin levels were downregulated in the PDAC cells by vitamin C (Figure 5B), and vitamin C treatment inhibited the transportation of β-catenin from the cytosol into the nucleus (Figure 5C). 

To further confirm the importance of Wnt signaling for vitamin C-mediated Snail regulation, we assessed the levels of Snail and EMT markers in the absence or presence of the Wnt activator, CHIR99021. As shown in Figure 5D, vitamin C treatment of the PDAC cells inhibited Snail expression, but these levels were significantly restored by exposure to CHIR9902 and the addition of CHIR99021 to the growth medium rescued the reduced migration following vitamin C treatment. Consistent with this result, the mesenchymal marker levels downregulated by vitamin C treatment were also robustly recovered by the addition of CHIR99021 to the growth medium (Figure 5E). We were further able to rescue the vitamin C treatment-induced inhibition of migration by activating Wnt signaling using Wnt3a-conditioned media obtained from Wnt3a-overexpressing L929 cells (Figure 5F). Collectively, these results indicated that vitamin C could inhibit metastasis through the inhibition of Wnt/β-catenin signaling.

## 3. Discussion

We here demonstrate that vitamin C inhibits glucose metabolism in PDAC cells, leading to the significant suppression of their growth. Furthermore, vitamin C was found in our present analyses to suppress Snail transcription factor expression, which in turn decreased PDAC metastasis via the inhibition of the Wnt/β-catenin signaling pathway.

Previous studies have reported that vitamin C has diverse functions in the body, such as enzyme activation, antioxidation and immune function [32]. By the 1970s, vitamin C had already been used as a cancer therapy by Pauling, who reported that at high doses, it had positive effects on survival in advanced cancer patients [33]. In addition, evidence has accumulated that the pharmacological concentration of vitamin C functions as a pro-oxidant and causes a buildup of hydrogen peroxide, which is directly cytotoxic against cancer cells [34]. High-dose vitamin C treatments were also found in another study to selectively kill some cancer cells by elevating the endogenous ROS level and causing DNA damage [35]. Reczek et al. demonstrated that a high-dose vitamin C exposure induced cell death through the uptake of its oxidized form, dehydroascorbate (DHA), into cells via the GLUT1. In cells, DHA is reduced to vitamin C by glutathione (GSH), which affects the maintenance of the level of GSH and nicotinamide adenine dinucleotide phosphate (NADPH) [36]. Elevated ROS levels in cancer cells have cytotoxic impacts. Furthermore, several studies have reported that vitamin C enhances the activity of specific gene transcription and epigenetic regulation through the activation of histone demethylase and ten-eleven translocation (TET) enzyme potentiality [37]. Indeed, various prior reports indicate that vitamin C-mediated epigenetic regulation improved the activation of chemotherapy against various cancers [13,17,38]. Our present results show that high-dose vitamin C exposure inhibits growth and activates apoptotic cell death in PDAC cells through the inhibition of glucose metabolism. This represents a potential new avenue for the development of pancreatic cancer therapies.

Cancer cells reprogram their metabolic pathways to increase their ATP production and macromolecule synthesis to levels that will support their proliferation requirements [39]. Glucose metabolism is important for providing a carbon source for ATP production and biomass that will enable cellular growth and proliferation. The aerobic glycolytic pathway is enhanced by the activation of oncoproteins such as KRAS and by the inhibition of tumor suppressor functions [2,40]. Under aerobic conditions, cancer cells strongly metabolize the conversion of glucose to lactic acid to retain mitochondrial respiration [41]. The disruption of mitochondrial respiration directly causes the death of cancer cells by enhancing their sensitivity to chemotherapeutic drugs [2]. As a consequence of this, mitochondrial respiration pathways have become an attractive target for cancer therapy development. A previous report has shown that vitamin C induces apoptotic death in gastric cancer cells via mitochondrial dysregulation [42]. In our present analyses, vitamin C was shown to mediate growth inhibition and apoptotic death in PDAC cells by inhibiting glucose metabolism. Significantly, the addition of pyruvate to the culture medium significantly rescued the cells from these effects. These findings support the notion that vitamin C may cause mitochondrial dysfunction. Yun et al. reported in this regard that vitamin C could selectively kill KRAS and BRAF mutant colorectal cancer cells via the inactivation of glyceraldehyde 3-phosphate dehydrogenase (GAPDH). Low levels of GSH are known to induce ROS accumulation, the inactivation of GAPDH, the inhibition of glycolysis and subsequent ATP depletion, resulting in cell death [21]. Although we did not observe any alteration in the ROS levels or expression of GAPDH in our present study, we have provided evidence from our current analyses that high-dose vitamin C promotes apoptotic cell death through mitochondrial dysfunction. We do not yet know the precise mechanism of glucose metabolism inhibition by vitamin C, and this awaits elucidation in the future.

One prominent feature of progressive cancers is their metastatic ability, which contributes to early-stage dissemination. EMT processes are central to this and are induced by various transcription factors such as Snail and Slug that inhibit E-cadherin and activate Vimentin expression [29]. Of note, in particular, the inhibition of E-cadherin expression plays an important role in EMT processes and is regulated by canonical Wnt signaling [30]. Emerging evidence has now demonstrated that Wnt/β-catenin signaling-mediated EMT regulation overcomes chemotherapy resistance and inhibits tumorigenesis in various cancers [43,44]. Moreover, previous studies have shown that high-dose vitamin C suppresses the invasiveness and metastatic capacity of certain cancer cells [12,13]. Consistent with previous results, we revealed in our current experiments that high-dose vitamin C inhibits the metastasis of PDAC cells and regulates the expression of major EMT factors. In the EMT process, the Snail transcription factor, one of the Wnt/β-catenin signaling targets, activates mesenchymal gene expression to induce the metastasis of PDAC cells [45,46]. Indeed, our present results have revealed that high-dose vitamin C downregulates Snail expression and inhibits metastasis via the suppression of Wnt/β-catenin signaling. Wnt signaling interferes with the phosphorylation of β-catenin and causes its cytosolic accumulation and subsequent nuclear translocation, resulting in the upregulation of Wnt target genes, including Snail [30]. Likewise, our current data indicated that high-dose vitamin C treatment suppresses the non-phospho β-catenin levels (Figure 5B). In addition, the inhibition of GSK3β led to a significant recovery of Snail and mesenchymal marker expression, which were reduced by high-dose vitamin C. Many studies have reported the utility of the Wnt pathway as a cancer therapy target because Wnt/Snail signaling plays a key role in cancer invasion and metastasis [47]. In addition, Lee et al. have described another function of Wnt/Snail signaling that causes mitochondrial dysfunction [48].

In summary, we here describe for the first time that a high-dose vitamin C treatment controls the expression of EMT markers through the regulation of Wnt/β-catenin signaling-mediated Snail expression. This may serve as an attractive therapeutic target for pancreatic cancer in the future.

## 4. Materials and Methods

### 4.1. Cell Culture and Reagents

The pancreatic cancer cells used in this study (8988T and 8902) were acquired from the American Type Culture Collection (ATCC, Manassas, VA). These cells were incubated at 37 °C in humidified air with 5% CO_2_ and grown in DMEM, and RPMI 1640 medium (Costar, Corning, NY, USA) supplemented with 10% fetal bovine serum (FBS), 100 U/mL penicillin, and 100 ug/mL streptomycin (Hyclone, Pittsburgh, PA, USA). All cells were routinely tested for mycoplasma contamination. Vitamin C was acquired from Sigma-Aldrich (St. Louis, MO, USA). CHIR99021 was obtained from MedChemExpress, and zVad-fmk (FMK001) was purchased from R&D systems (R&D systems, Minneapolis, MN, USA).

### 4.2. Cell Growth and Cell Viability Assays

For the growth assay, cells were plated in 24-well plates (density: 2000 cells/well), and the medium was not changed during the course of the experiment. On the indicated days, the cells were fixed in 10% formalin, washed with DPBS, and then stained with 0.1% crystal violet. The dye was extracted with 10% acetic acid, and measurements were then taken at 595 nm using a plate reader. Cell viability was determined using an MTT assay. Briefly, the cells were seeded in 96-well plates at a density of 2000 cells/well. After 10 days, the medium was changed and supplemented with 100 µL MTT solution (0.5 mg/mL). The cells were then incubated at 37 °C for 3 h. The reaction was stopped by replacing the MTT solution with 100 µL DMSO. The formazan salts were dissolved by shaking incubation for 5 min. The solubilized formazan was measured at 570/630 nm.

### 4.3. Colony Formation Assay

Cells were plated in 6-well plates at a density of 600–800 cells/well. The medium was not changed throughout the experiment. After 10 days, the cells were fixed in 80% methanol and stained with a 0.2% crystal violet.

### 4.4. Annexin V/PI Assay

Cells were seeded in 12-well plates at a density of 0.8 × 10^5^ cells/well. Cells were then cultured in the medium under different conditions for 24 h, harvested by trypsinization, washed with DPBS, and resuspended in 1X Annexin-V/PI binding buffer (10 mM HEPES, 140 mM NaCl, 2.5 mM CaCl_2_, pH 7.4) containing Annexin-V FITC and propidium iodide (PI). Cells were then quantified and analyzed by flow cytometry (Beckman-Coulter, Brea, CA, USA).

### 4.5. Western Blot

For western blotting, equal amounts of extracted protein from the cells were resolved by sodium dodecyl sulfate-polyacrylamide gel electrophoresis (SDS-PAGE) and transferred to a polyvinylidene difluoride membrane (PVDF). The membranes were blocked with 5% degreased milk in Tris-buffered saline and 0.1% Tween-20 (TBST) and incubated overnight with the appropriate primary antibody against PARP, caspase-3, Snail, n-cadherin, zo-1, non-phospho-β-catenin (Cell Signaling Technology, MA, USA), vimentin, E-cadherin or β-actin (Santa Cruz Biotechnology, Santa Cruz, CA, USA). The following day, membranes were incubated with appropriate secondary antibodies conjugated with HRP. The signals on the blots were detected using an enhanced chemiluminescence (ECL) solution.

### 4.6. Metabolomics

Cells were plated in 10-cm dishes at a density of 2 × 10^6^ cells/dish in triplicate. The following day, the cells were washed several times with PBS and water and then collected with 1.4 mL of methanol/H_2_O (80/20, *v*/*v*). A 100 µL internal standard (5 µM) was then added to the preparation and dissolved by vigorous vortexing. Metabolites were extracted from the aqueous phase after the addition of chloroform. These extracts were dried using a vacuum centrifuge and dissolved in 50 µL of 50% methanol prior to LC-MS/MS analysis.

### 4.7. Extracellular Acidification Rate Measurement

The cells were seeded into a 24-well Seahorse plate (Seahorse Bioscience, North Billerica, MA) and incubated at 37 °C in humidified air with 5% CO_2_. The medium was replaced with unbuffered DMEM, and the cells were then further incubated at 37 °C in humidified air without CO_2_ for 1 h. Glucose, oligomycin and 2-deoxy-D-glucose (2DG) were then added to final concentrations of 10, 1 and 20 mM, respectively. The extracellular acidification rate (ECAR) in each sample was then measured on an XF24 extracellular flux analyzer (Seahorse Bioscience).

### 4.8. Quantification of Intracellular ATP

Intracellular ATP concentrations were determined using an ATP Colorimetric/Fluorometric Assay kit (K354-100, BioVision, Milpitas, CA, USA) in accordance with the supplier’s instructions. After 1 × 10^6^ cells were plated, the cells were exposed to different conditions through a medium change for 24 h. The cells were harvested by scraping, washed with DPBS, and lysed in 100 µL ATP assay buffer. Aliquots containing 50 µL of supernatant were added to a 96-well plate with 50 µL ATP assay buffer containing an ATP probe, ATP converter, and developer. The absorbances of these preparations were then measured at 570 nm.

### 4.9. Transwell Assay for Invasion and Migration

Invasion and migration assays were conducted using a Transwell system (Corning). For the invasion assay, the 8988T and 8902 cells were seeded at a density of 3 × 10^5^ cells/well in 100 µL of serum-free DMEM medium in the inner membrane coated with Matrigel (BD Biosciences, Mountain View, MA) of the upper chamber. DMEM containing 10% FBS (750 µL) was added to the lower chamber. For the migration assay, 30,000 cells of 8988T and 8902 cells were seeded in 200 µL of serum-free DMEM medium in the inner membrane coated with 0.4% gelatin of the upper chamber, and the complete medium was added to the lower chambers. After incubation of the cells for 48 h, the cells on the top surface of the inner membrane were removed by cleaning with a cotton swab. The cells were then fixed in 10% formalin and stained with 0.5% crystal violet. The values representing the degree of invasion and migration were averaged from three independent experiments.

### 4.10. TOF/FOP Promoter Assay

One day prior to their transfection, the cells were seeded at 12-well plates at a density of 3 × 10^5^ cells/well. The cells were transfected on the following day using PEI with 0.1 µg of TOP flash (T-cell factor reporter plasmid) or FOP flash (negative control with mutant T-cell factor reporter binding sites) and 1.8 µg of pCMV-β-galactosidase (β-gal) in serum-free media. The cells were then incubated at 37 °C for 6 h with serum-free medium and then changed to the complete medium. At 48 h later, the cells were collected and lysed in RIPA lysis buffer with a protease inhibitor cocktail. The luciferase activities of TOP flash and FOP flash were measured using a Luciferase Assay System (E1501; Promega, Madison, WI, USA).

### 4.11. Statistics

The data in this study are presented as a mean value ± standard deviation. All statistical comparisons were performed using an unpaired Student’s t-test.

## 5. Conclusions

This study provides reliable evidence for the antitumor activity of vitamin C against PDAC. High-dose vitamin C treatments were found to suppress the growth of PDAC cells and promote their apoptotic cell death by inhibiting glucose metabolism, thus leading to a disruption of mitochondrial function. Furthermore, through the inhibition of Wnt/β-catenin signaling, vitamin C was observed to inhibit Snail-mediated EMT marker expression, which in turn suppresses metastasis. Vitamin C is, therefore, an attractive potential therapeutic agent for pancreatic cancer.

## Figures and Tables

**Figure 1 ijms-23-12249-f001:**
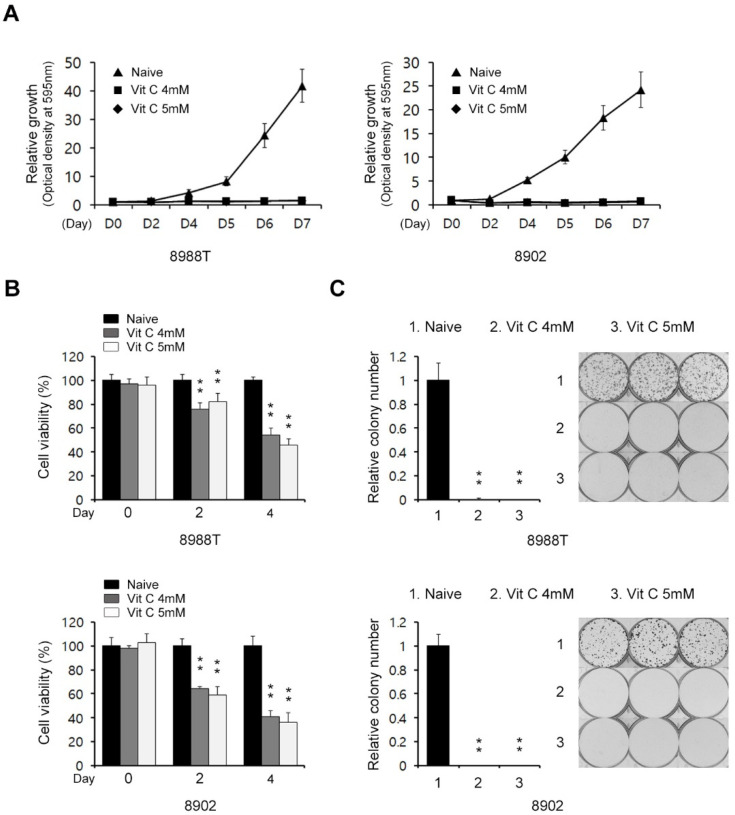
Effects of high-dose vitamin C on PDAC cell growth. (**A**) Growth assay for PDAC cells treated with vitamin C (4 or 5 mM) for the indicated days. (**B**) Cell viability assay for PDAC cells treated with vitamin C (4 or 5 mM) for 24 h. (**C**) Clonogenic assay for PDAC cells treated with vitamin C (4 or 5 mM) for 8 days. Error bars represent the s.d. of triplicate wells from representative experiments; ** *p* < 0.01.

**Figure 2 ijms-23-12249-f002:**
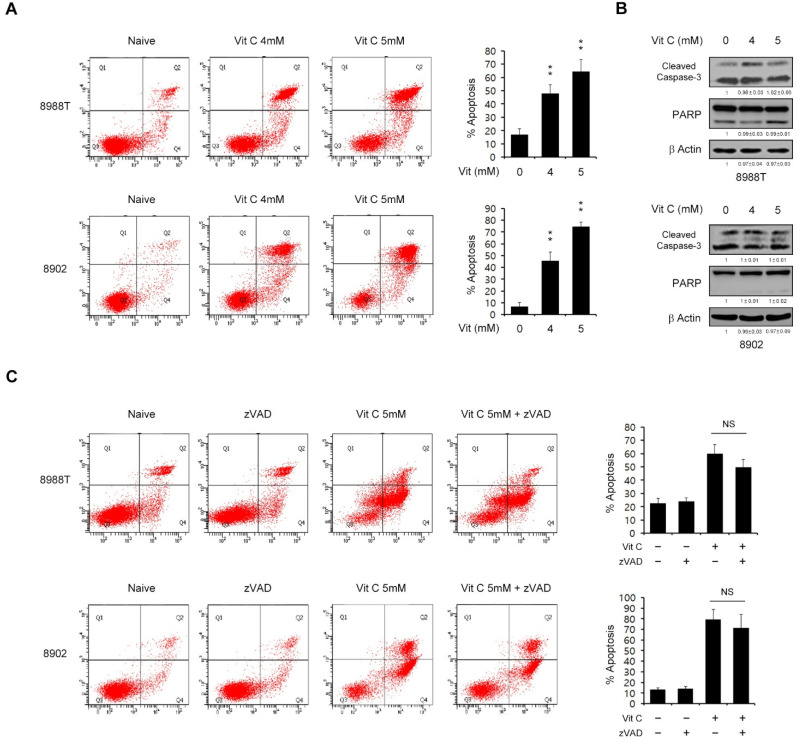
High-dose vitamin C enhances the caspase-independent apoptosis response of PDAC cells. (**A**) PDAC cells were treated with vitamin C (4 or 5 mM) for 24 h and assayed for apoptotic cell death by annexin V/PI staining and flow cytometry. (**B**) PDAC cells were treated with vitamin C (4 or 5 mM) for 24 h and immunoblotted with the indicated antibodies. Error bars with the mean ± s.d. of three independent experiments. (**C**) PDAC cells were treated with vitamin C (4 or 5 mM) for 24 h with or without zVAD-fmk (50 μM) and assayed for apoptotic cell death by annexin V/PI staining and flow cytometry. Error bars represent the s.d. of triplicate wells from representative experiments; ** *p* < 0.01. NS, not significant.

**Figure 3 ijms-23-12249-f003:**
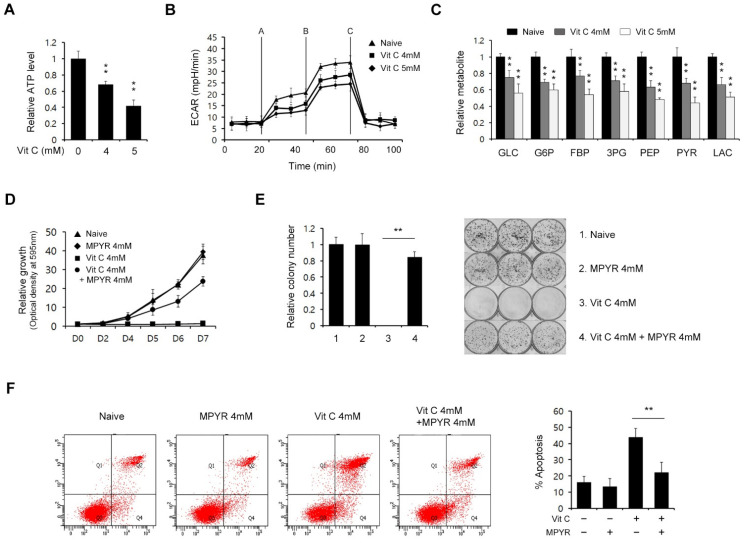
High-dose vitamin C suppresses the glycolytic flux in PDAC cells. (**A**) 8988T cells were treated with vitamin C (4 or 5 mM) for 24 h and assayed for intracellular ATP. (**B**) 8988T cells were treated with vitamin C (4 or 5 mM) for 24 h, and the extracellular acidification rate was measured. (**C**) 8988T cells were treated with vitamin C (4 or 5 mM) for 24 h and analyzed for glycolysis metabolite pools via LC/MS-MS. (**D**) Cell growth assay for 8988T cells treated with vitamin C (4 mM) for 8 days with or without methyl-pyruvate (4 mM). (**E**) Clonogenic assay for 8988T cells treated with vitamin C (4 mM) for 8 days with or without methyl-pyruvate (4 mM). Error bars represent the s.d. of triplicate wells from a representative experiment. (**F**) 8988T cells were treated with vitamin C (4 mM) for 8 days with or without methyl-pyruvate (4 mM) and assayed for apoptotic cell death by annexin V/PI staining and flow cytometry. Error bars represent the s.d. of triplicate wells from representative experiments; *** p* < 0.01.

**Figure 4 ijms-23-12249-f004:**
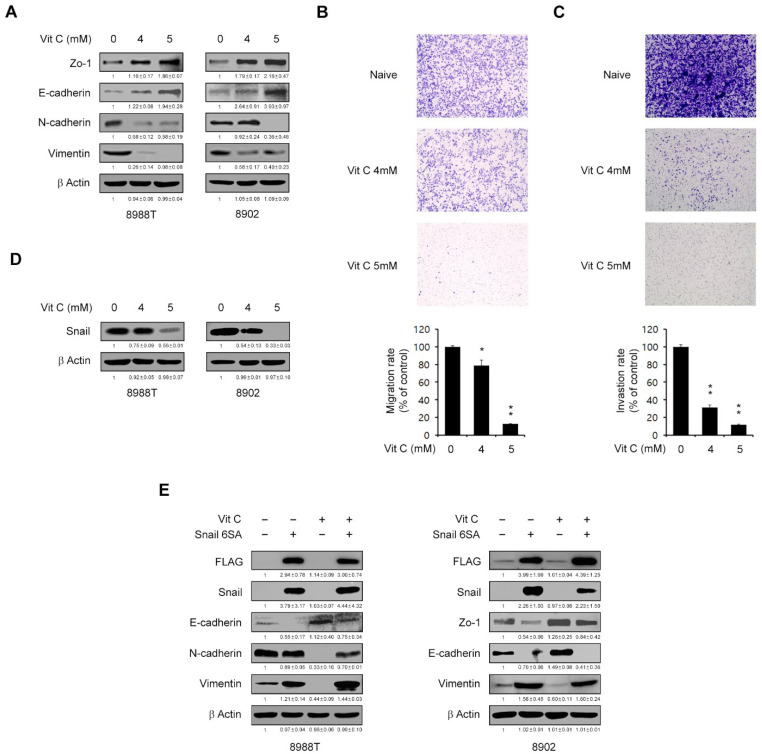
High-dose vitamin C suppresses the invasiveness and metastasis of PDAC cells. (**A**) PDAC cells were treated with vitamin C (4 or 5 mM) for 24 h and immunoblotted with the indicated antibodies. Error bars with the mean ± s.d. of three independent experiments. (**B**) Migration assay for 8988T cells loaded into the upper chamber of a 0.4% gelatin-coated transwell and treated with vitamin C (4 or 5 mM) for 24 h in the lower chamber. The bar plot represents the percentage of cell migration for each panel. (**C**) Invasion assay for 8988T cells seeded into the upper chamber of a Matrigel (1 mg/mL) coated transwell and treated with vitamin C (4 or 5 mM) for 24 h in the lower chamber. The bar plot represents the percentage of cell invasion for each panel. (**D**) PDAC cells were treated with vitamin C (4 or 5 mM) for 24 h and immunoblotted with the indicated antibodies. Error bars with the mean ± s.d. of three independent experiments. (**E**) PDAC cells expressing FLAG-Snail 6SA were treated with vitamin C (5 mM) for 24 h and immunoblotted with the indicated antibodies. Error bars reflect the s.d. of three independent experiments; * *p* < 0.05, ** *p* < 0.01.

**Figure 5 ijms-23-12249-f005:**
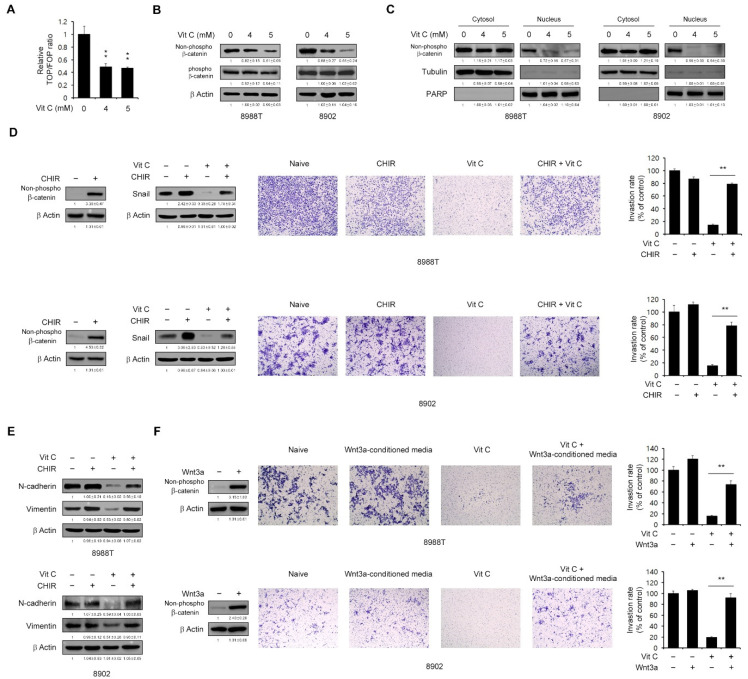
Effects of vitamin C on the migration ability of PDAC cells. (**A**) TOP/FOP luciferase reporter assays of 8988T cells treated with vitamin C (4 or 5 mM) for 24 h. (**B**,**C**) PDAC cells were treated with vitamin C (4 or 5 mM) for 24 h and immunoblotted with the indicated antibodies. Error bars reflect the s.d. of three independent experiments. (**D**,**E**) PDAC cells were treated with vitamin C (5 mM) for 24 h with or without CHIR and immunoblotted against the indicated antibodies. Invasion assay for PDAC cells seeded into the upper chamber of a Matrigel (1 mg/mL) coated transwell and treated with vitamin C (5 mM) for 24 h in the lower chamber. The bar plot represents the percentage of cell invasion for each panel. (**F**) Invasion assay for PDAC cells seeded into the upper chamber of a Matrigel (1 mg/mL) coated transwell and treated with vitamin C (5 mM) for 24 h in the lower chamber. Error bars represent the s.d. of triplicate wells from a representative experiment among three independent experiments; *** p* < 0.01. CHIR, CHIR99021.

## Data Availability

The data that support the present results are available from the corresponding author upon reasonable request.

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
