# Peer review of "Vitamin C Suppresses Pancreatic Carcinogenesis through the Inhibition of Both Glucose Metabolism and Wnt Signaling"

_ijms, 2022, doi:10.3390/ijms232012249_

Round 1

Reviewer 1 Report

The manuscript by Kim et al investigated the anti-cancer effects of vitamin C against pancreatic ductal adenocarcinoma (PDAC) cells and its mechanisms through glycolysis and Wnt signaling. Authors in this article also focused on effects of vitamin C on as epithelial-mesenchymal transition (EMT) and invasion of PDAC. However, overall, it seems that the verification in this article is insufficient and below are some of the major concerns.

Major comments

1.     Overall, authors didn't describe the number of biological independent experiments. Therefore, data shown in this article is unreliable. Furthermore, authors showed only representative images of the western blot. Authors should quantify the data of western blot and perform statistical analysis. For the same reason, the invasion assay should be performed quantitation and statistical analysis.

2.     Glycolysis predominates in cancer cells, this is known as The Warburg effect. In other words, cancer cell is dependent on glucose metabolism to obtain ATP rapidly. As author knows, ATPs are synthesized from 1,3-BPG and PEP in glycolysis pathway. This ATP-synthesis process is before pyruvate production. In Figure 3, vitamin c treatment reduced ATP synthesis, indicating that vitamin C inhibited early step of glycolysis pathway and induced apoptosis of PDAC. In this article, the authors showed that administration of pyruvate restored the effect of vitamin C. Reviewer couldn't understand why pyruvate restored several phenotypes by treatment of vitamin C because pyruvate cannot provide ATP from itself.

Minor comments

1.     The authors have not fully explained why authors came up with the idea of being involved in the Vitamin C and WNT signaling in introduction part.

2.     In Fig. 3B, What is “measurement number”?

3.     Author didn't describe what is “Snail 6SA” in Fig. 4E. Please explain this.

4.     Author described “The Wnt/β-catenin signaling pathway is  … the abnormal regulation of the β-catenin transcription factor. “ in line 176. β-catenin is not a transcription factor. Please correct this.

Reviewer 2 Report

Major comments:

1.     What is the maximum concentration of Vit C in human uptake? How many percentages of Vit C can be uptake by tumor cells after injection or eating.  

2.     If possible, please also investigate the inhibitory effect of Vit C (>4 mM) in a normal cell line in Figure 1. Or please explain why high-dose Vit C does not affect normal cells?

3.     What is the mechanism for Vit C suppressing glycolysis? Via competition with glucose (GLUTs) or Wnt-beta-Catenin axis or any other specific receptor protein of Vit C in tumor cells or just induction of cellular stress such as ROS production?

4.     What is the mechanism for Vit C suppressing Wnt-beta-catenin axis? If possible, please also detect phosphorylated beta-catenin in Figure 5B.

Reviewer 3 Report

Kim et al. have reported that high dose of vitamin C impaired the growth of PDAC cells by inhibiting glucose metabolism. The authors have also showed that vitamin C inhibited the invasion of PDAC cells by Snail and Wnt signaling-mediated EMT. Although most of the results in this study were demonstrated clearly with robust effects, there are still some major concerns for further consideration. In addition, novelty is lacking in this study.

1. The authors need to demonstrate the dose-dependent effect of vitamin C in Figure 1 instead of starting with a high concentration which killed all cells. Thus, further results from this high concentration of vitamin C treated cells are not convincing. 

2. The two cell lines used here need characterization. In Figure 3, the results need to be confirmed in at least two cell lines.

3. In Figure 5, the location of β-catenin needs to be shown to confirm activity.

4. The selectivity of the Wnt activator needs to be demonstrated. The wnt3a-overexpressing L929 cells need confirmation. In Figure 5C and E, the effects of treatments of wnt3a-CM and CHIR on EMT need to be proved by both WB and invasion assay in multiple cell lines.

5. Metastasis ability can't be concluded from the invasion assay. More experiments need to be performed to support the conclusion that vitamin C inhibits metastasis.

6. In general, some of the figures lack clear label of which cell line and the manuscript might benefit from language editing. 

Round 2

Reviewer 1 Report

Author revised along to reviewer's comment well. However, reviewer would like to mention about below.

Reviewer’s comment to Response 2

The author explained that the administration of pyruvate can upregulate OXPHOS in Vitamin C-treated PDAC. However, there is no data indicating this hypothesis. Author should confirm OXPHOS in above condition measured by XF24 (oxygen consumption rate, OCR). If author’s explanation is true, author should describe this theory in result part.

Reviewer 2 Report

No more questions. This study is ready for publication.

Author Response

We deeply appreciate the quality of the review and the constructive comments. We believe that the insights of the referee and our subsequent revisions have greatly enhanced our study.

Reviewer 3 Report

The authors have addressed most of the concerns. It would be better to include the figure in the response letter as a supplementary figure and discuss the results in the manuscript, so that it will be clear how the concentration of vitamin C was chosen. 
